# Impact of Chitosan Nanoparticles-Coated Dendritic Cell-Based Vaccine as Cancer Immunotherapy

**DOI:** 10.3390/vaccines13050474

**Published:** 2025-04-28

**Authors:** Jehan S. Alrahimi, Najla S. Alotaibi, Alia M. Aldahlawi, Fatemah S. Basingab, Kawther A. Zaher

**Affiliations:** 1Department of Biological Sciences, Faculty of Science, King Abdulaziz University, Jeddah 21589, Saudi Arabia; 2Immunology Unit, King Fahd Medical Research Center, King Abdulaziz University, Jeddah 21589, Saudi Arabia; 3Department of Medical Laboratory Sciences, Faculty of Applied Medical Sciences, King Abdulaziz University, Jeddah 21589, Saudi Arabia

**Keywords:** dendritic cells, chitosan-nanoparticles, tumor microenvironment, antitumor vaccine, immune response, drug-delivery system

## Abstract

Dendritic cells (DCs) are major contributors to generating an effective immune response due to their ability to present antigens to T cells. Recently, nanoparticles have been widely used in different medical applications, such as drug-delivery systems, to enhance the function of impaired immune cells. **Objectives:** This research aims to develop an effective antitumor DC-based vaccine by adsorption of chitosan-nanoparticles (CH-NPs) onto DCs. **Methods:** Undifferentiated mouse bone marrow progenitor cells were differentiated into mature DCs using cytokines and lipopolysaccharides. CH-NPs were prepared using the ionic gelation method and subsequently used to coat the stimulated DCs. The MTT assay was employed to assess the cytotoxicity of all formulations. To compare the antitumor effect of CH-NPs, DCs, and DCs-CH-NPs, mice were divided into five groups and injected with the respective vaccine formulations. Following immunization, flow cytometry was used to analyze DC and CD4^+^ T cell activation in blood and spleen tissues. Histological samples from the spleen and lymph nodes were also collected. **Results:** Our findings show that co-stimulatory molecules CD80/CD86 and the DC maturation marker CD83 were upregulated in the vaccinated DCs, indicating their maturation. Moreover, CD83, CD11c, and MHC-II were upregulated in blood and spleen samples in vivo. The DC-CH-NPs vaccinated group had a higher mean percentage of CD83 expression in blood samples (76.7 ± 17.1) compared to the DCs group (47.7 ± 11.0) and the CH-NPs group (37.7 ± 8.6). DC markers, particularly CD83, were highly expressed in spleen samples. Additionally, the DC-CH-NPs vaccinated group had a significantly higher number of CD4^+^ T cells (MFI = 26.1 ± 2.3) compared to the DCs (18.6 ± 1.6) and CH-NPs (13.3 ± 1.4) groups. **Conclusions:** The present study concludes that the DC-CH-NPs vaccine formulation can induce a potent in vivo immune response. These data may provide valuable insights for developing effective delivery systems for antitumor vaccines.

## 1. Introduction

Cancer immunotherapy, a type of biotherapy, is considered a promising non-conventional treatment, especially for non-curable and metastatic cancers [1]. In cancer, immunotherapies can recognize [2] and eliminate tumor cells through the patient’s immune system without exhibiting cytotoxicity or harming healthy cells and tissues, as occurs with chemotherapy and radiotherapy. During immunotherapy, immune cells identify and destroy tumor cells using several strategies, including T cell-based immunotherapy and vaccines that activate cytotoxic T lymphocytes (CTL) [2].

DCs are a unique type of immune cell that circulate in the blood and are located in most tissues and lymphoid organs, such as the skin, inner nose, lungs, stomach, and intestines [3]. These cells regulate the functions of both T and B cells and enhance adaptive immune responses against specific antigens [4]. Given their potent ability to process and present antigens, DCs hold promise for treating various advanced diseases, including autoimmune disorders and cancer, and preventing transplant rejection [5,6,7,8]. However, tumors may interfere with DC function, as the tumor microenvironment (TME) contains immunosuppressive cytokines and factors, such as IL-6 and GM-CSF, that impair DC maturation [9]. This results in an increased population of immature DCs and a reduction in mature DCs, weakening the immune response against tumors. Several studies have shown that DC-based vaccines can effectively eliminate tumor cells by inducing antigen-specific immune responses [10,11]. Furthermore, combining DC-based vaccines with conventional therapies, such as chemotherapy, has been suggested to improve efficacy [12].

Nanotechnology applications are emerging as promising tools in immunotherapy due to their ability to enhance antitumor immune responses. Nanoparticles can interact with the immune system to direct and deliver immune cells, initiating an immune response against tumor cells. Due to their small size, nanoparticles are well-known carriers for improving drug efficacy, as they can easily penetrate the tumor vasculature. Consequently, these features allow for targeted delivery to specific cells such as T cells, DCs [13], and/or tumor cells [14,15].

Nevertheless, DC-based vaccines face several challenges related to their effectiveness. El-Sissi et al. [16] developed a chitosan nanoparticle (CH-NP) vaccine to address these limitations and enhance immune responses against tumors. In 2015, Hunsawong and colleagues reported a novel dengue nanovaccine in which CH-NPs were loaded with cell wall components from *Mycobacterium bovis* Bacillus Calmette-Guerin. This approach functioned as an adjuvant to deliver the vaccine and stimulate immune responses by promoting DC maturation and cytokine release [17]. Thus, nanoparticles are increasingly used in immunotherapy as delivery vehicles. Consequently, many studies aim to develop nanoparticle-coated vaccines to ensure efficient delivery [18]. The present research aims to develop an effective antitumor DC-based vaccine using CH-NPs to activate CD4^+^ T immune responses.

## 2. Materials and Methods

### 2.1. Preparation and Characterization of CH-NPs

#### 2.1.1. Preparation of Chitosan Nanoparticles

Ionotropic gelation with Tripolyphosphate (TPP) was used to prepare CH-NPs, as described in [19]. Briefly, a 1% acetic acid aqueous solution was used to dissolve medium molecular weight chitosan powder (200–800 cP). A 0.125% (1 mg/mL) TPP solution was prepared using double-distilled water. Rapid mixing was employed to achieve smaller nanoparticles. The formed CH-NPs were designed to remain in a nanoparticle suspension to ensure effective intramuscular vaccination. The mixture was filtered through a 0.22 µm filter to obtain a final chitosan stock solution at 0.1% (*w*/*v*, 1 mg/mL). These positively charged CH-NPs are designed to interact with dendritic cell membranes primarily through electrostatic attraction, as the negative surface charge of cell membranes facilitates nanoparticle adsorption.

#### 2.1.2. Characterization of Chitosan Nanoparticles

The size distribution and zeta potential of CH-NPs were assessed using a Zetasizer Nano-ZS90 (Malvern Instruments, Worcestershire, UK). The presence of chitosan functional groups in the CH-NPs solution was evaluated using Fourier-transform infrared (FT-IR) spectroscopy, as previously described in [20]. Samples were diluted with deionized distilled water at various intensity concentrations and analyzed at a scattering angle of 90° at 25 °C. The structural characterization of chitosan and CH-NPs was performed using FT-IR spectroscopy (Shimadzu Scientific Instruments, Kyoto, Japan) to confirm the presence of chitosan functional groups, as previously described in [20].

### 2.2. Cell Lines and Cell Culture

#### 2.2.1. Cell Culture Preparation

The human breast cancer cell line MCF-7 (passage 14) was used to assess the toxicity of CH-NPs. The King Fahd Medical Research Center (KFMRC), King Abdulaziz University, Jeddah, Saudi Arabia, kindly provided the cells. The cells were incubated at 37 °C in a humidified atmosphere containing 5% CO_2_ using a HERACELL 240i incubator (Thermo Scientific, Waltham, MA, USA).

#### 2.2.2. Cell Culture Treatment

A 1 mg/mL stock solution of CH-NPs was diluted in cell culture media to obtain various concentrations. MCF-7 cells were treated with CH-NPs at concentrations of 1000, 500, 250, 125, and 62.5 μg/mL. As a control, cells were treated with 0.1% DMSO. All treatments were incubated for 24 h at 37 °C in an atmosphere of 5% CO_2_.

### 2.3. Generation of Mouse Bone Marrow Dendritic Cells

#### 2.3.1. Ethical Approval

The Animal Care and Use Committee (ACUC) at the King Fahd Medical Research Centre (KFMRC) reviewed and approved the study protocol (Approval #ACUC-20-12-39). All animal experiments involving mice were conducted according to the guidelines and regulations of the National Bioethical Committee of Saudi Arabia (NACSA), under the oversight of King Abdulaziz City for Science and Technology (KACST), Jeddah, Saudi Arabia.

#### 2.3.2. Experimental Protocol

Six female BALB/c albino mice (n = 6; age: 6 to 8 weeks) were used for producing and phenotyping bone marrow dendritic cells (BMDCs), with two animals per experiment across three experiments. The mice were housed in the Animal Facility at KFMRC under Specific Pathogen-Free (SPF) conditions. After euthanasia, the femurs were collected, and the bone marrow was flushed using a needle attached to a 1 mL syringe filled with cold, complete RPMI-1640 medium. The bone marrow suspension was passed through a cell strainer into a 15 mL conical tube to remove debris and then centrifuged at 300× *g* for 10 min at 4 °C to pellet the cells.

BMDCs were generated using the method described in [21]. Briefly, the bone marrow suspension was washed twice with PBS at 300× *g* for 10 min at 4 °C, resuspended in 5–10 mL of Ammonium-Chloride-Potassium (ACK) lysing buffer, and incubated at room temperature for 3–5 min to lyse red blood cells. After counting using an automated cell counter (Thermo Fisher, Waltham, MA, USA), cells were adjusted to a concentration of 2 × 10^6^ cells/mL and cultured in complete RPMI-1640 medium supplemented with recombinant murine GM-CSF (20 ng/mL; R&D Systems, Lausanne, Switzerland; Catalog #415-ML-020/CF) and IL-4 (10 ng/mL; BioLegend, San Diego, CA, USA; Catalog #574304). Cells were incubated at 37 °C in a 5% CO_2_ atmosphere. Media were partially replaced on days 3 and 5 with fresh cytokine-containing media. Before harvesting, DCs were stimulated with 1 µg/mL lipopolysaccharide (LPS) for 24 h.

#### 2.3.3. Flow Cytometry for Dendritic Cell Morphology and Phenotype

Flow cytometry (Beckman Coulter Life Sciences, Brea, CA, USA) was used to characterize the phenotypes of both immature and mature DCs. After seven days, cells were collected in 50 mL tubes, centrifuged at 300× *g* for 5 min at 4 °C, and washed three times with ice-cold staining buffer (PBS + 1% FBS). Cells were counted and stained in the dark for 30 min on ice with the following fluorochrome-labeled antibodies (1 µg/2 × 10^6^ cells): anti-mouse CD86-FITC (Catalog #105109), CD80-FITC (Catalog #104705), CD83-FITC (Catalog #121505), and CD14-FITC (Catalog #123307) (BioLegend, San Diego, CA, USA). After staining, cells were washed twice with cold FACS buffer and analyzed using flow cytometry (Beckman Coulter Life Sciences, USA), with 10,000 events recorded per sample.

### 2.4. Adsorption of Chitosan Nanoparticles onto Dendritic Cells

CH-NPs were incubated with mature DCs to facilitate non-covalent surface adsorption, primarily through electrostatic interactions between the positively charged nanoparticles and the negatively charged dendritic cell membranes. The effects of various CH-NP concentrations on dendritic cell viability were investigated. The MTT assay [3-(4,5-dimethylthiazol-2-yl)-2,5-diphenyltetrazolium bromide] was utilized to assess the safety of CH-NPs for dendritic cells. The half-maximal inhibitory concentration (IC_50_) was calculated to determine the CH-NP concentration that inhibits DC survival by 50%. Following the manufacturer’s guidelines, the MTT assay was conducted using the CyQUANT™ MTT Cell Viability Assay (Thermo Fisher, Catalog #V13154, Waltham, MA, USA).

### 2.5. In Vivo Assessment

#### 2.5.1. Animal Model

Thirty female BALB/c mice (average weight = 19.58 ± 0.19 g; age: 6 to 8 weeks) were obtained from the Animal House and maintained at KFMRC under pathogen-free conditions. Mice were housed in wire-bottomed cages with unrestricted access to food and water. Environmental conditions were controlled at 22 °C (±2), with 40–60% humidity and a 12-h light/dark cycle.

#### 2.5.2. Experimental Study Design

The mice were divided into five groups (n = 6), and each mouse was assigned a unique identification number. The study compared DCs, CH-NPs, and DC-CH-NPs as vaccine formulations. Groups I and II served as control groups. Groups III, IV, and V were vaccinated with DCs, CH-NPs, and DC-CH-NPs. Experimental outcomes were measured through body weight during the study and spleen weight at the end. On days 1, 14, and 21, Groups III–V received three intramuscular vaccine doses in the left flank. On day 7, mice were injected subcutaneously with syngeneic breast cancer cells in the mammary area. Group II (positive control) received tumor cells without treatment. Group III received CH-NPs (46.5 μg), Group IV received 1 × 10^5^ DCs in 100 μL PBS, and Group V received 1 × 10^5^ DCs associated with 46.5 μg of CH-NPs in 100 μL PBS after washing (Figure 1).

#### 2.5.3. Assessing DC and T Cell Activation

On day 35, the mice were euthanized. Flow cytometry evaluated DC maturation and T cell activation in blood and spleen tissues. Cells were isolated by centrifugation at 300× *g* for 5 min at 4 °C and washed twice with FACS buffer. They were stained with FITC-labeled antibodies against CD4, CD83, CD14, CD11c, and MHC-II. Additional spleens were preserved for histological analysis.

#### 2.5.4. Histological Studies

Spleens from the negative control group were removed, weighed, and compared with those from the positive control and vaccinated groups. Spleen and lymph node tissues were collected from all groups immediately after sacrifice on day 35. Tissues were prepared as per the protocol in [22], sliced to 1–2 mm thickness, and fixed in 10% paraformaldehyde for 24 h. Following fixation, tissues were dehydrated using graded alcohol concentrations, cleared in xylene for 30 min, and embedded in paraffin wax. Thin sections (4–5 μm) were cut using a microtome. Sections were stained with hematoxylin for 10 min, rinsed with water, counterstained with eosin for 1 min, and dehydrated. Finally, slides were cleared in three xylene baths, air-dried, and mounted with coverslips for microscopic examination.

### 2.6. Statistical Analysis

The mean and standard error of the mean (SEM) were used to summarize continuous variables. Data from the treatment groups were analyzed using one-way analysis of variance (ANOVA), followed by Tukey’s post-hoc test and Dunnett’s test for multiple comparisons. The statistical difference between mature and immature dendritic cells was assessed using a two-sample *t*-test. Data analysis and visualization were conducted using RStudio (Version 4.1.2) and Microsoft Excel 2019. Visualizations were generated using the “ggplot2” and “ggpubr” packages in R. A *p*-value of less than 0.05 was considered statistically significant.

## 3. Results

### 3.1. Morphology of Differentiated Bone Marrow-Derived Dendritic Cells

Throughout the bone marrow cell culture period, morphological changes were monitored using an inverted microscope equipped with digital imaging software F 4.00.00 software (Nikon Eclipse Ti, Nikon Instruments Inc., Fujisawa, Japan). On day 1, the cells appeared small, round, and firmly adherent to the culture plate surface (Figure 2A). Starting from day 2, the cells began detaching, aggregating, and forming both large and small clusters of bone marrow-derived dendritic cells (BMDCs) (Figure 2B–D). On day 7, the differentiated cells were stimulated with lipopolysaccharide (LPS). By day 8, distinct morphological differences between immature and mature DCs were observed under a phase-contrast inverted microscope (Figure 2E,F). Immature DCs displayed short cytoplasmic projections (Figure 2E), whereas mature DCs exhibited elongated or irregular shapes with multiple, long dendritic projections (Figure 2F).

### 3.2. Dendritic Cell Phenotype by Flow Cytometry

Flow cytometric analysis was carried out using forward and side scatter gating, as shown in Figure 3A. Mature DCs were characterized by surface markers of high expression of co-stimulatory molecules, CD80, CD86, and CD83, and low or lack of expression of monocyte markers, CD14. Percentages of DC surface markers varied between LPS-treated and untreated DCs (Figure 3B). In addition, DC surface marker expression was assessed using flow cytometry and presented as mean fluorescence intensity (MFI) (Figure 3C).

The results showed that the expression of CD14 in LPS-treated DCs was significantly downregulated (MFI = 181.9 ± 2) compared to untreated DCs (MFI = 506.6 ± 4.7) (*p* < 0.001). CD14 is a monocyte marker that occasionally decreases while monocytes differentiate into DCs. At the same time, the expression of membrane-bound CD83 is increased on the surface of activated DCs and is considered a major marker used to distinguish between mature and immature DCs. In the current study, the maturation marker CD83 expression was significantly upregulated (MFI = 522.2 ± 3.6) in mature DCs compared to unstimulated immature DCs (MFI = 125.9 ± 5.3) (*p* < 0.01). Furthermore, co-stimulatory molecules CD80 and CD86 are expressed by APCs such as mature DCs; those markers are crucial in initiating the signal to activate T cells. Both CD80 and CD86 markers were highly expressed in mature DCs at MFI = 235.3 ± 6.6 and 329.6 ± 4.7, respectively, in comparison to immature DCs at MFI = 154.6 ± 3.2 and 222.9 ± 1.2, (*p* < 0.001, 0.02, respectively). Data from several experiments indicated that the DCs were fully matured after stimulation with LPS (Table 1).

### 3.3. Size, Zeta Potential, and Morphology of Chitosan Nanoparticles

The UV spectrum of the prepared CH-NPs showed a maximum wavelength at 200 nm, corresponding to the peak absorption of chitosan. The particle size distribution ranged from 1 to 100 nm, qualifying them as nanoparticles. The zeta potential of the CH-NPs was +26.1 ± 0.8 mV, confirming the positively charged surface and indicating good suspension stability (Figure 4A).

Fourier transform infrared (FT-IR) spectroscopy (Nicolet 6700 FT-IR, Thermo Fisher Scientific, Waltham, MA, USA) was used to verify the chemical structure and presence of functional groups in both chitosan and CH-NPs. This step was essential to confirm nanoparticle formation. FT-IR analysis revealed peaks at 3321 cm^−1^ corresponding to the –OH and –NH_2_ stretching vibrations, indicating the functional integrity of the particles. FT-IR spectroscopy was used to confirm the presence of functional groups characteristic of chitosan. While the observed –NH_2_ and –OH peaks are consistent with chitosan, they are not unique indicators of nanoparticle formation and are commonly present in both chitosan and TPP-crosslinked formulations. Thus, FT-IR alone does not confirm nanoparticle synthesis or DC interaction and should be interpreted accordingly.

Two primary peaks were observed in the CH-NPs spectrum: one near 3300 cm^−1^ (–OH) and another around 1600 cm^−1^ (–NH_2_). The NH_2_ bending vibrations typically appear between 1500–1700 cm^−1^. The mean particle size was measured at 65 nm, with a polydispersity index (PDI) of 0.224, indicating a moderately heterogeneous population. While some smaller particles (~3 nm) may appear in the distribution, these likely represent background or solvent-related signals and are not representative of the CH-NPs population (Figure 4B).

### 3.4. Cytotoxicity Study Using MTT Assay

A significant difference was observed among CH-NP concentrations (one-way ANOVA, *p* < 0.001). A gradual decrease in MCF-7 cell viability was noted with increasing concentrations of CH-NPs. Concentrations below 15 μg/mL maintained cell viability above 90%. The control group differed significantly from all concentrations, except for the lower concentrations of 2, 4, and 8 μg/mL (multiple comparisons, *p* < 0.05). The 15 μg/mL concentration was marginally significant (*p* = 0.049), suggesting no substantial difference in cytotoxicity compared to lower concentrations. When comparing cell viability between CH-NPs, DC-CH-NPs, and controls, a significant difference in cell viability percentages was found between the control and treatment groups at all concentrations except 4 and 2 μg/mL.

### 3.5. In Vivo Assessment

#### 3.5.1. Tumor Induction and Vaccination Effect on Body Weight

Body weight was monitored every 7 days. There were no significant differences in body weight among the vaccinated groups (repeated measures ANOVA, *p* = 0.437). Mice in the negative control group exhibited a steady weight gain, reaching 20.25 g by the end of the experiment, while the positive control group showed the lowest weight (20.10 g). The DC-CH-NPs, DC, and CH-NPs groups had final weights of 20.22 g, 20.20 g, and 20.17 g, respectively. These findings suggest that DCs associated with CH-NPs helped maintain weight post-tumor induction.

#### 3.5.2. Assessing Dendritic Cell Activation in Mice Blood

Flow cytometric analysis of DC surface markers in blood revealed that CD83 expression was significantly upregulated in the DC-CH-NPs group (76.7 ± 17.1%, *p* = 0.012) compared to the positive control group. The DC and CH-NPs groups also showed increased CD83 expression (47.7 ± 11% and 37.7 ± 8.6%), though not statistically significant (*p* = 0.171 and *p* = 0.476, respectively). CD14 expression was also significantly elevated in the DC-CH-NPs group (74.0 ± 18.8%, *p* = 0.025). Notably, CD11c and MHC-II expression were significantly higher in the DC group compared to the positive control (48.3 ± 11% and 58.7 ± 2.6%, *p* = 0.02), and CD11c expression was even higher in the DC-CH-NPs group (94.1 ± 6.1%, *p* = 0.006). MHC-II expression levels were nearly equal in the DC and DC-CH-NPs groups (71.7 ± 3.4%, *p* = 0.02) (Figure 5A,B).

#### 3.5.3. Assessing Dendritic Cell Activation in Mice Spleen

In the spleen, CD83 expression was significantly elevated in the DC-CH-NPs group (87.2 ± 14%, *p* = 0.01), followed by the DC group (65.7 ± 10.5%, *p* = 0.07) when compared to the positive control group. The CH-NPs group showed the lowest CD83 expression (40.2 ± 2.1%, *p* = 0.84).

CD14 expression was highest in the positive control group (71.6 ± 5.1%), followed by DC-CH-NPs (70.8 ± 3.7%), DCs (47.4 ± 12.7%), and CH-NPs (26.7 ± 9.4%). CD11c and MHC-II expression were significantly increased in the DC-CH-NPs group (90.8 ± 11.9% and 91.2 ± 10.8%, *p* = 0.02 and 0.005, respectively) compared to the other vaccinated groups (Figure 5C,D).

#### 3.5.4. Assessing T Cell Activation

T cell activation was evaluated by analyzing CD4^+^ T cell marker expression. The most CD4^+^ T helper cells were observed in the DC-CH-NPs group across both blood and spleen samples. Blood samples from all groups were stained with FITC-anti-CD4 and analyzed using FCS Express 7 software (De Novo Software, Kitchener, Ontario, Canada) (Figure 6A).

The mean fluorescence intensity (MFI) of CD4^+^ T cells in blood differed significantly across groups (one-way ANOVA, *p* < 0.001). All treatment groups showed significantly higher MFI values than the negative control group (Tukey post-hoc test, *p* < 0.001). The CH-NPs group (MFI = 12.52 ± 2.1) and the positive control group (MFI = 13.34 ± 1.6) did not differ significantly (*p* = 0.614), indicating limited activation. MFI values were significantly higher in the DC-CH-NPs group (78.20 ± 1.4), followed by the DC (46.14 ± 1.3) and CH-NPs (26.52 ± 1.5) groups (*p* < 0.001 for all comparisons). The spleen showed higher overall MFI values than blood across all groups (Figure 6B,D).

### 3.6. Histological Studies of Spleen and Lymph Nodes of Vaccinated Mice Groups

#### 3.6.1. Histological Findings for Spleen

The spleen plays a central role in regulating blood cell composition, including platelets, red blood cells (RBCs), and white blood cells (WBCs). It is divided into two central structural regions: red pulp (RBP) and white pulp (WLP). The RBP comprises venous sinuses, splenic cords, and a filtration system for old, damaged, or unnecessary RBCs. Additionally, it contains WBCs that eliminate pathogens such as viruses, bacteria, and fungi and acts as a reservoir for WBCs and platelets. The WLP is involved in producing and maturing B and T lymphocytes and generating antibodies. The marginal zone, which separates the WLP from the RBP, is key in filtering bloodborne pathogens into the WLP. These compartments are typically well-defined (Figure 7A).

In the positive control group, histological examination of the spleen revealed several pathological changes due to tumor induction, including tumor cell infiltration and necrosis. The RBP and WLP appeared expanded, and their structural boundaries were less distinct. The normal architecture of the germinal center was disrupted, indicating splenomegaly. Anomalies in the central artery (CA) were noted, including multiple arteries within the pulp (Figure 7B).

In the CH-NPs-vaccinated group, normal spleen architecture was altered. The RBP, WLP, and marginal zone were not well delineated. There was a noticeable reduction in WLP cellularity, and the CA wall appeared thickened (Figure 7C). The DC-vaccinated group showed similar histological characteristics to the CH-NPs group. In particular, a fusion of the RBP, loss of distinct pulp boundaries, and absence of a recognizable marginal zone were evident. A thickened CA wall was also observed (Figure 7D).

In contrast, the DC-CH-NPs-vaccinated group showed more preserved spleen architecture. The RBP and WLP were more distinguishable, and the marginal zone was identified (Figure 7E). Interestingly, a higher concentration of white cells was observed in the RBP, with abundant red blood cells. The CA endothelial cells appeared compressed, likely due to high cellular pressure from erythrocytes, leukocytes, and cellular debris, including platelets (Figure 7E).

#### 3.6.2. Histological Findings of the Lymph Nodes

Lymph nodes are key immune system components responsible for filtering and removing pathogens that invade the body. These small, bean-shaped organs contain distinct populations of B and T lymphocytes, which mediate immune surveillance and response. Structurally, each lymph node comprises two regions: the capsule and the parenchyma. The capsule is a fibrous outer layer penetrated by afferent lymphatic vessels. Beneath it lies the subcapsular sinus, a space facilitating the entry and movement of lymph fluid.

The parenchyma consists of the cortex and the medulla. The cortex, located beneath the subcapsular sinus, contains naive B and T cells and antigen-presenting cells such as dendritic cells and macrophages. It is further subdivided into the outer cortex, rich in B lymphocytes, and the inner cortex or paracortex, which predominantly houses T lymphocytes. These zones surround the medulla, where active plasma cells reside and contribute to antibody production (Figure 8A).

In the positive control group (PBS-treated mice injected with tumor cells), histological analysis revealed significant pathological alterations, including tumor infiltration into the lymph node parenchyma, disorganized lymphatic nodules, and the absence of germinal centers. Additionally, megakaryocytes, blast cells, and polymorphonuclear cells, ordinarily absent, were observed. These findings suggest severe immune depletion, possibly followed by compensatory extramedullary hematopoiesis and lymphocyte regeneration (Figure 8B).

The CH-NPs-vaccinated group exhibited non-circular, naked lymphoid follicles with irregular germinal centers (Figure 8C). The DC-vaccinated group showed reduced follicular density; in some cases, tumor infiltration into the lymph node parenchyma was evident (Figure 8D). Notably, the DC-CH-NPs-vaccinated group presented an accumulation of lymphocytes in the paracortex (Figure 8E) along with the increased presence of follicular dendritic cells (FDCs), macrophages, and polymorphs—indicative of enhanced lymphoid activity and potential antigen presentation (Figure 8E).

## 4. Discussion

Breast cancer remains one of the most prevalent malignancies worldwide, primarily affecting women, though it also occurs occasionally in men [23]. Pursuing safe, targeted, and effective cancer therapies has intensified interest in immunotherapeutic approaches, particularly DC-based vaccines. Due to their biocompatibility, non-toxicity, and ability to encapsulate bioactive molecules, chitosan nanoparticles (CH-NPs) have emerged as a valuable tool in drug delivery systems, including immunotherapies [18,24,25,26]. Their unique physicochemical properties make them ideal carriers for enhancing antigen presentation and stimulating adaptive immune responses. This study aimed to synthesize CH-NPs using the ionic gelation method, adsorb them onto in vitro-differentiated DCs, and assess their cytotoxicity and immunological efficacy in a murine breast cancer model.

DCs were successfully isolated from mouse bone marrow and cultured using granulocyte-macrophage colony-stimulating factor (GM-CSF) and interleukin-4 (IL-4), both essential for their differentiation and survival [27]. IL-4 enhances DC maturation from progenitor cells [28,29,30]. By day 8, the DCs exhibited hallmark morphological features, including dendritic projections, and were further matured using lipopolysaccharide (LPS), a potent stimulator of toll-like receptor 4 (TLR4) that triggers downstream signaling cascades and enhances co-stimulatory molecule expression [27].

Phenotypic analysis confirmed successful DC maturation, characterized by the upregulation of CD80, CD86, and CD83 surface markers, along with decreased CD14 expression—hallmarks of functional antigen-presenting cells [31,32]. Interestingly, CD86 expression exceeded CD80, aligning with previous reports showing that CD86 is rapidly expressed upon stimulation, whereas CD80 requires more sustained signaling for surface expression [33,34]. Tumor necrosis factor (TNF) has also been implicated in the upregulation of CD80 [34,35].

CH-NPs are widely studied in drug delivery research due to their biocompatibility and mucoadhesive properties [19]. CH-NPs were synthesized via the ionic gelation method, which offers advantages over traditional techniques by avoiding harsh conditions that may denature proteins [36,37,38]. The CH-NPs produced had a narrow size distribution under 100 nm and a zeta potential of +26.1 ± 0.8 mV, indicating good suspension stability and efficient interaction with cells, consistent with previous studies [39,40]. FTIR analysis confirmed the presence of characteristic functional groups, including –NH_2_ and –OH, supporting successful nanoparticle formation and compatibility for DC adsorption. While the FT-IR spectrum confirms the presence of functional groups typical of chitosan, it is not by itself conclusive evidence of nanoparticle formation. Therefore, we have supplemented our interpretation by including mean particle size and PDI from dynamic light scattering (DLS) measurements, which confirm the formation of CH-NPs with a predominant size of 65 nm (PDI = 0.224).

The MTT assay demonstrated that CH-NPs were safe at concentrations below 300 µg/mL, maintaining more than 50% DC viability. A 465.1 µg/mL concentration was selected for effective DC adsorption, and no significant cytotoxic differences were observed between CH-NPs and DC-CH-NPs groups, supporting the biosafety of the final formulation [37]. These results emphasize that smaller nanoparticles with well-defined physicochemical properties facilitate cellular uptake via passive endocytosis mechanisms.

To evaluate in vivo immunogenicity, vaccinated mice were assessed for DC maturation and T-cell activation in blood and spleen tissues. Flow cytometry showed significant upregulation of CD83, CD11c, and MHC-II markers in the DC-CH-NPs group, confirming enhanced DC activation [41,42]. CD14 expression was markedly reduced, indicating successful differentiation and maturation. These findings underscore the synergistic potential of combining DCs with CH-NPs to amplify immune responses.

Moreover, CD83 plays a central role in stabilizing MHC-II expression on thymic epithelial cells and is essential for CD4^+^ T cell selection [43,44]. The transmembrane domain of CD83 modulates its internalization and degradation while promoting the expression of MHC-II and CD86 on activated antigen-presenting cells, including DCs and B cells [41,45]. The increased CD83 expression observed in this study highlights its critical role in T cell priming and adaptive immunity.

Elevated CD14 expression observed in some cases may reflect the presence of CD14^+^ DC subsets, which possess macrophage-like properties that are inclined toward inducing humoral immunity [46,47]. These CD14^+^ cells resemble murine CD11b^+^ dermal dendritic cells and may contribute to immune modulation depending on the tumor microenvironment.

Although CD8^+^ T cells play a crucial role in tumor immunity by directly targeting and eliminating malignant cells [48], this study did not assess CD8^+^ T cell responses. The primary objective was to evaluate the effectiveness of chitosan nanoparticle-associated dendritic cells (DC-CH-NPs) in activating CD4^+^ T helper cells and promoting dendritic cell maturation, serving as a preliminary investigation into the vaccine’s immunogenicity. Several practical considerations contributed to the exclusion of CD8^+^ T cell profiling, including limited availability of flow cytometry reagents, panel design restrictions, and the strategic decision to initially focus on early-stage immune activation events mediated by DC–CD4^+^ T cell interactions. It is also important to note that CD4^+^ T cells play a pivotal role in coordinating adaptive immune responses, including facilitating CD8^+^ T cell priming, enhancing B cell activity, and supporting cytokine production [49]. Future studies will expand the immunophenotyping panel to include CD8^+^ T cell analysis and functional assessments, such as granzyme B, perforin, and IFN-γ expression, which are essential for fully characterizing the cytotoxic arm of vaccine-induced antitumor immunity.

The DC-CH-NP formulation demonstrated superior immunological activity compared to CH-NPs or DCs alone. This platform enhanced DC maturation, antigen presentation, and CD4^+^ T cell activation while maintaining biosafety and tissue integrity. Integrating nanotechnology and immunotherapy presents a promising strategy for cancer vaccine development and may be further optimized for clinical translation.

## 5. Conclusions

In conclusion, this research aimed to evaluate the impact of DC-CH-NPs vaccine formulation in breast cancer-induced mice. Using the cytokines GM-CSF and IL-4, dendritic cells (DCs) were successfully generated and differentiated in vitro from bone marrow-derived cells. Concurrently, chitosan nanoparticles (CH-NPs) were synthesized and evaluated for their stability and biological effectiveness. Following confirmation of full DC maturation through morphological and phenotypic characterization, DC-CH-NPs were formulated. Cytotoxicity was assessed using the MTT assay to determine the safe concentration of CH-NPs and the optimal DC dose for vaccine formulation.

An in vivo study was conducted in a murine breast cancer model to compare the immunological impact of DC-CH-NPs, DCs alone, and CH-NPs. Flow cytometric analysis of blood and lymph node samples revealed that the DC-CH-NPs-vaccinated group exhibited a higher expression of the DC maturation marker CD83 (76.7 ± 17.1%) compared to the DCs group (47.7 ± 11.0%) and CH-NPs group (37.7 ± 8.6%).

Additionally, DC markers were detected in spleen samples, particularly CD83. The DC-CH-NPs-vaccinated group also showed a significantly higher number of CD4^+^ T cells in blood samples (MFI = 26.1 ± 2.3) compared to the DCs (18.6 ± 1.6) and CH-NPs (13.3 ± 1.4) groups. A notably stronger T-cell response was also observed in spleen tissues from the DC-CH-NPs group (MFI = 78.20 ±1.4). Histopathological examination of spleen and lymph node sections stained with H&E confirmed a robust immune response in the DC-CH-NPs group.

These findings provide compelling evidence that the DC-CH-NPs formulation effectively stimulates an immune response in vivo. The outcomes of this study suggest the potential of this formulation as a novel vaccine delivery strategy to enhance immuno-therapeutic approaches for cancer prevention and treatment. However, further studies are warranted to validate these findings and expand immunological assessments, including the involvement of CD8^+^ T cells, to elucidate the formulation’s role in adaptive antitumor immunity fully.

## Figures and Tables

**Figure 1 vaccines-13-00474-f001:**
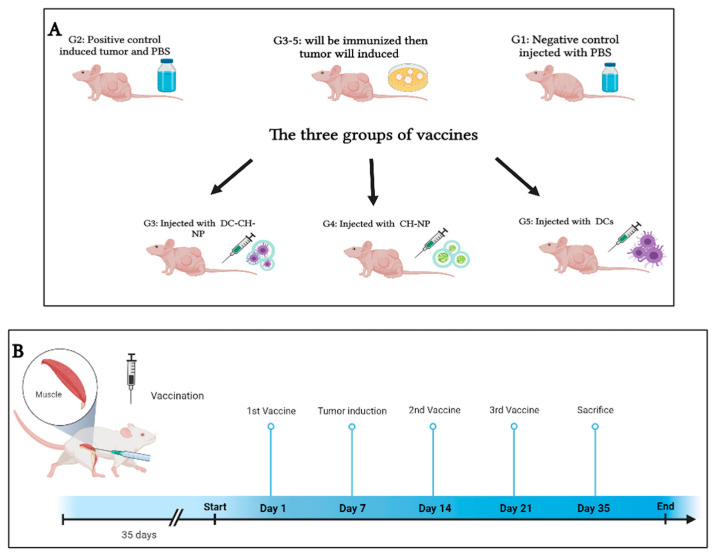
Overview of the vaccination study design. (**A**). The diagram shows the steps of the mouse vaccination and tumor challenge. (**B**). The schedule of immunization and tumor challenge.

**Figure 2 vaccines-13-00474-f002:**
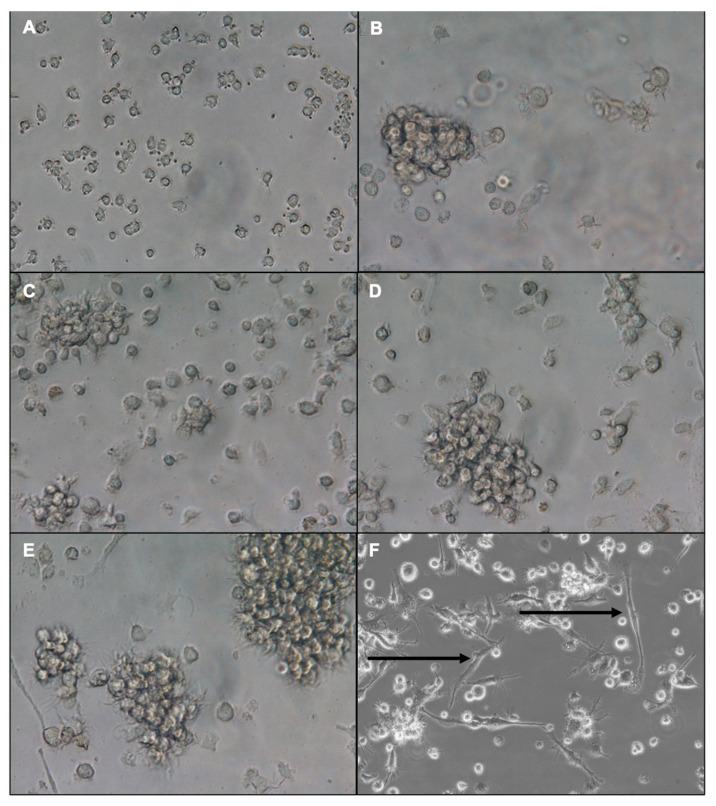
Morphological differentiation of mouse bone marrow-derived dendritic cells during culture. BM cells were cultured for seven days in the presence of GM-CSF and IL-4. On day 1, the cells appeared small and round (**A**). From days 2 to 6, cells began aggregating and forming clusters (**B**–**D**). On day 7, unstimulated DCs had irregular shapes and short cytoplasmic projections (**E**), while LPS-stimulated mature DCs were elongated and irregular, with numerous long projections (**F**). Images were captured with an inverted microscope at 20× and 40× magnification at the Immunology Unit, KFMRC, King Abdulaziz University. Arrows in (**F**) highlight the dendritic projections.

**Figure 3 vaccines-13-00474-f003:**
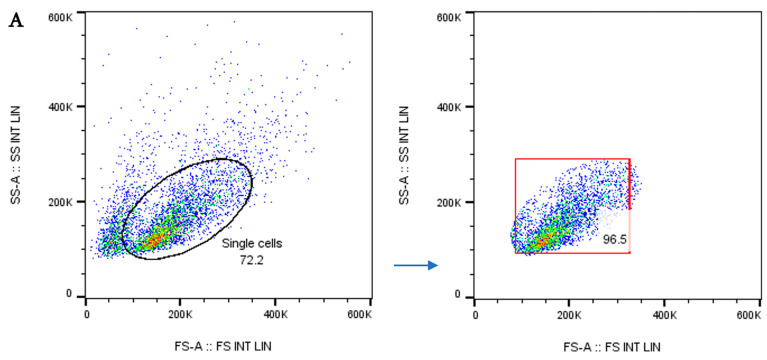
(**A**) Forward and side scatter gating strategy using FlowJo LLC (Version 10.7.1). (**B**) Phenotypically compared untreated and LPS-treated DCs using flow cytometry, showing CD83, CD80, CD86, and CD14 expression. (**C**) Mean fluorescence intensity (MFI) bar graphs of DC surface markers (CD14, CD80, CD83, and CD86) from four independent experiments. Data were analyzed using a *t*-test. Plots created using R version 4.2.1.

**Figure 4 vaccines-13-00474-f004:**
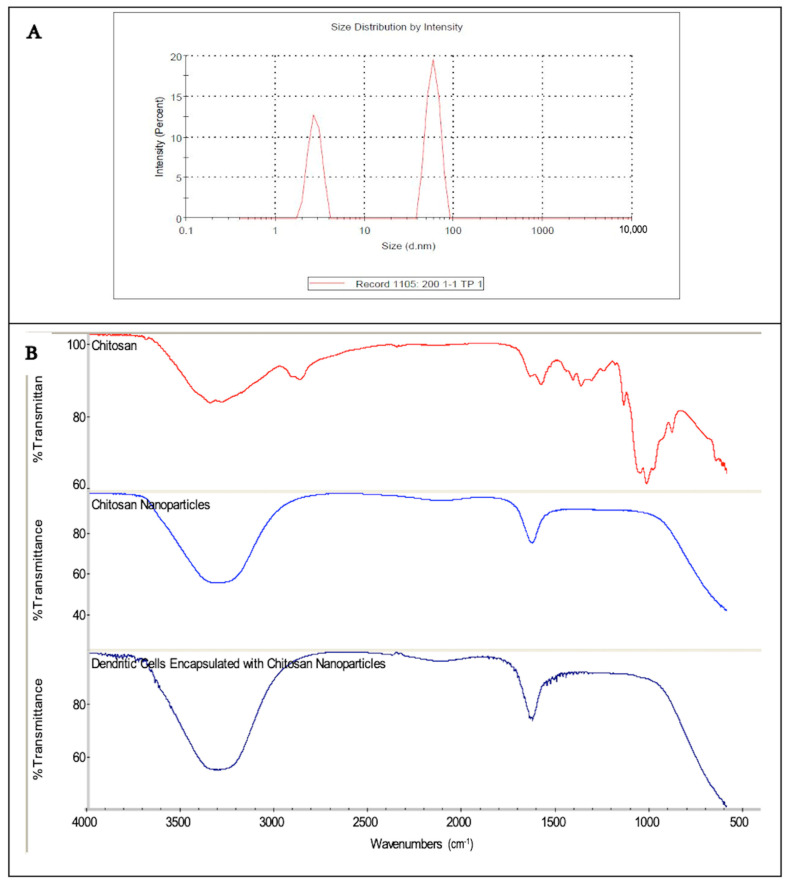
(**A**) Size distribution of CH-NPs. The average particle size was 65 nm, with a 56–100 nm range. (**B**) FT-IR spectra showing functional group peaks for CH, CH-NPs, and CH-NP-associated DCs. (**B**) FT-IR spectra showing functional group peaks for CH, CH-NPs, and CH-NP-associated DCs. The figure shows a bimodal particle size distribution, with a primary nanoparticle population centered around 60–65 nm and a minor peak near 3 nm, possibly due to solvent background. Mean particle size = 65 nm, PDI = 0.224.

**Figure 5 vaccines-13-00474-f005:**
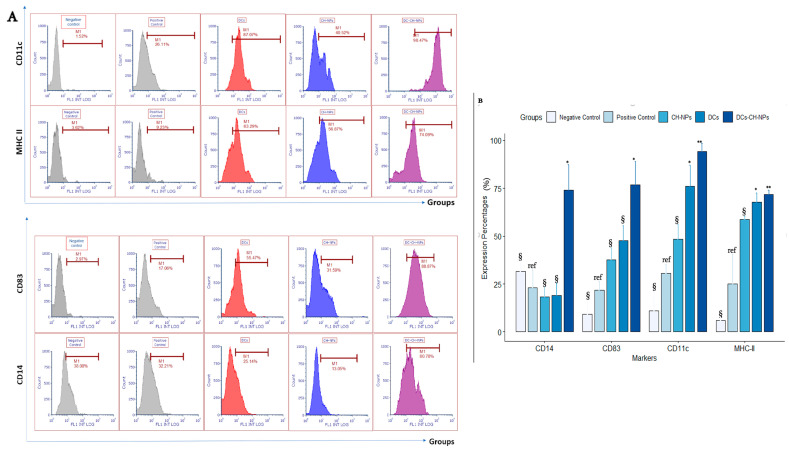
Dendritic cell analysis from blood and spleen samples. (**A**). Flow cytometric analysis of dendritic cell surface markers from blood samples. The first row shows an expression of the maturation marker CD83; the second row shows an expression of the monocyte marker CD14; the third row shows CD11c, a widely used marker for dendritic cells; and the fourth row shows the activation marker MHC-II. Data were analyzed using FCS Express 7 software (De Novo Software, Kitchener, ON, Canada). (**B**). Bar plots showing the mean expression percentages of dendritic cell markers in mouse blood samples following vaccination. Marker expression was compared among the vaccinated and positive control groups (the negative control group received PBS; the positive control group was injected with cancer cells and saline). Data were analyzed using one-way ANOVA (α = 0.05) followed by Dunnett’s post-hoc test. Significance levels: * *p* < 0.05; ** *p* < 0.01. “ref” denotes reference groups; “§” indicates non-significant differences. Results are presented as mean ± SEM. Plots were generated using R version 4.2.1. (**C**). Representative flow cytometric analysis of dendritic cell surface markers from spleen samples. Rows correspond to CD83, CD14, CD11c, and MHC-II, respectively. (**D**). Bar plots show dendritic cell markers’ mean expression percentages in mouse spleen samples after vaccination. Data were analyzed using one-way ANOVA (α = 0.05) followed by Dunnett’s post-hoc test. Significance levels: * *p* < 0.05; ** *p* < 0.01. “ref” denotes reference groups; “§” indicates non-significant differences. Results are presented as mean ± SEM. Plots were generated using R version 4.2.1.

**Figure 6 vaccines-13-00474-f006:**
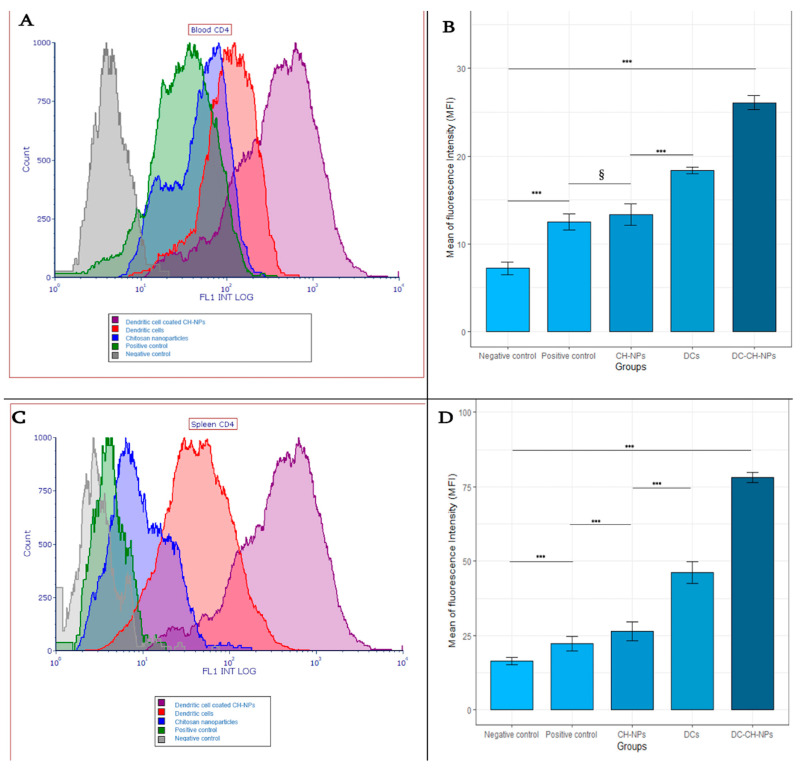
Flow cytometric analysis of CD4^+^ T Lymphocyte activation in mice. (**A**). Representative flow cytometric analysis of CD4^+^ T cell expression in blood samples collected from all experimental groups. Samples were stained with FITC-conjugated anti-CD4 antibody. Data were analyzed using FCS Express 7 software (De Novo Software, Ontario, Canada). (**B**). Bar plot of mean fluorescence intensity (MFI) values for CD4^+^ T cells in mouse blood samples following vaccination. A significant increase in MFI was observed in all treatment groups compared to the negative control group (one-way ANOVA, *p* < 0.001). Significance levels: *** *p* < 0.001. “§” indicates non-significant results. Data are presented as mean ± SEM. Plots were generated using R version 4.2.1. (**C**). Representative flow cytometric analysis of CD4^+^ T cells from spleen tissue. The spleen samples were stained with FITC-conjugated anti-CD4 antibody to evaluate T cell activation. Data were analyzed using FCS Express 7 software. (**D**). Bar plot of MFI values for CD4^+^ T cells in spleen samples. The highest MFI was observed in the DC-CH-NPs group, followed by the DCs and CH-NPs groups. Overall, CD4^+^ T cell activation was higher in the spleen tissue than in the blood. Statistical analysis was performed using one-way ANOVA with Tukey’s post-hoc test (α = 0.05). Significance levels: *** *p* < 0.001. Data are presented as mean ± SEM.

**Figure 7 vaccines-13-00474-f007:**
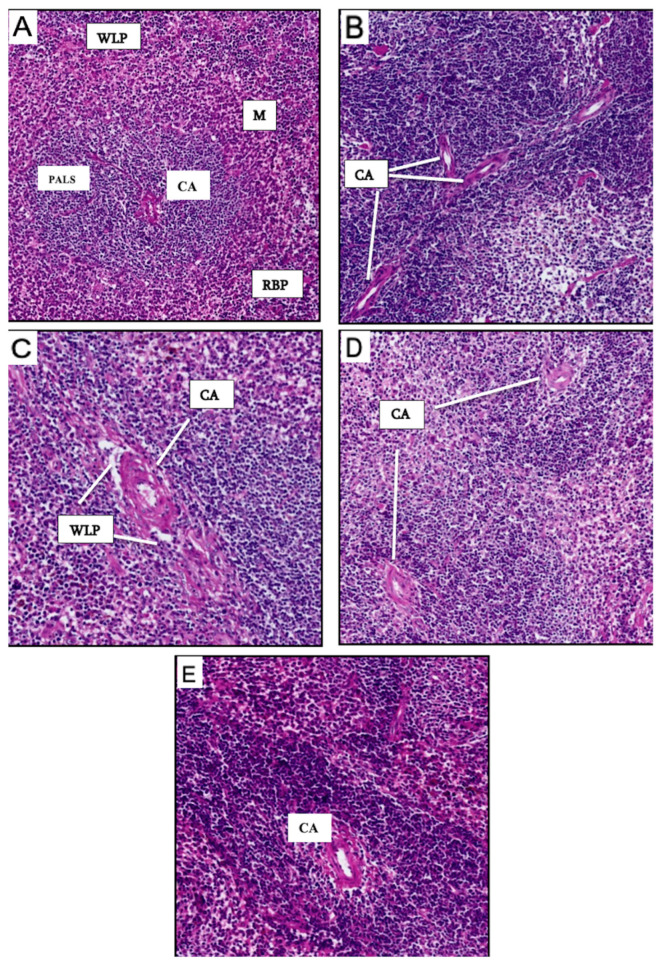
Histological analysis of spleen tissues from vaccinated and control mice. (**A**) Spleen section from the negative control group showing well-defined compartments: white pulp (WLP), red pulp (RBP), marginal zone (M), and a centrally located artery (CA) surrounded by a perivascular lymphoid sheath (PALS). (**B**) Positive control group spleen exhibiting pathological changes due to cancer induction, including thickened CA walls and the presence of multiple arteries in the pulp, suggesting disorganized architecture and splenomegaly. (**C**) Spleen from the CH-NPs vaccinated group, showing loss of structural definition between WLP and RBP, absence of a clear marginal zone, depleted WLP cellularity, and thickened CA wall. (**D**) The DC-vaccinated group showed similar histological changes as the CH-NPs group, including indistinct pulps, thickened CA walls, and fused RBP. (**E**) The DC-CH-NPs vaccinated group displayed improved structural integrity of the spleen, with distinguishable WLP, RBP, and marginal zones. Numerous white cells were present in the RBP, along with dense populations of erythrocytes and visible debris. The CA endothelial layer appeared compressed, likely due to high cell pressure. All spleen sections were stained with hematoxylin and eosin (H&E) and scanned using a Grundium Ocus scanner (Grundium, Tampere, Finland). Magnification: 10× (pixel resolution: 100 μm).

**Figure 8 vaccines-13-00474-f008:**
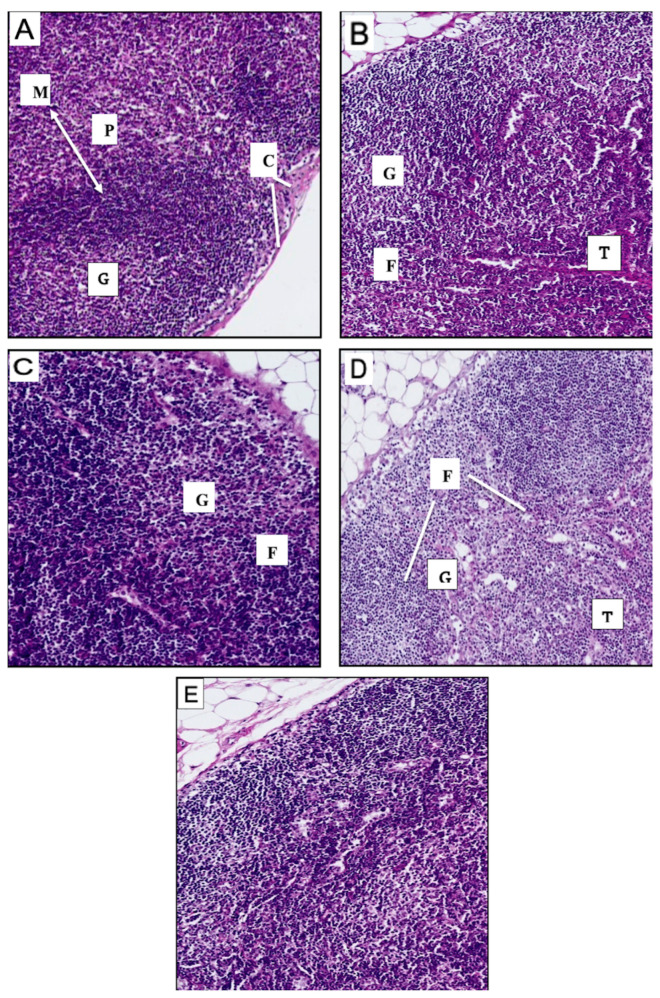
Histological analysis of lymph nodes from experimental mouse groups. (**A**) Lymph node from the negative control group showing a well-defined capsule, subcapsular sinus (C), cortex with lymphatic nodules (F), a rounded germinal center (G), paracortex (P), and medulla (M). (**B**) Positive control group lymph node infiltrated by tumor cells (T), with diffuse lymphatic nodules (F) and degenerated germinal center (G). (**C**): Lymph node from CH-NPs-vaccinated mice displaying irregular, non-circular germinal centers (G) and naked lymphoid follicles (F). (**D**): Lymph node from DC-vaccinated mice showing low follicular density, tumor cell infiltration (T), and reduced structural definition of lymphatic nodules (F) and germinal centers (G). (**E**): Lymph node from DC-CH-NPs-vaccinated mice exhibiting a dense accumulation of paracortical lymphocytes. All tissue sections were stained with hematoxylin and eosin (H&E) and scanned using a Grundium Ocus scanner (Grundium, Tampere, Finland). Magnification: 10× (100 μm resolution) for (**A**–**D**) 40× (50 μm resolution) for (**E**).

**Table 1 vaccines-13-00474-t001:** Phenotypic comparison of immature and mature DCs by MFI.

Type of DCs	+LPS Treated DCs (MFI)	−LPS Treated DCs (MFI)	^a^*p* Value
Markers			
CD14	181.9 ± 2.8	506.6 ± 4.7	0.001 ***
CD83	522.2 ± 3.6	125.9 ± 5.3	0.01 **
CD80	235.3 ± 6.6	154.6 ± 3.2	0.001 ***
CD86	329.6 ± 4.7	222.9 ± 1.2	0.02 *

^a^*p* value was based on two-sample *t*-test; * *p* < 0.05, ** *p* < 0.01, *** *p* < 0.001. MFI values were expressed as mean ± SD from four independent experiments.

## Data Availability

The data supporting this study’s findings are available upon request from the corresponding author. No new data were created.

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
