# Peer review of "Impact of Chitosan Nanoparticles-Coated Dendritic Cell-Based Vaccine as Cancer Immunotherapy"

_vaccines, 2025, doi:10.3390/vaccines13050474_

Round 1
Reviewer 1 Report
Comments and Suggestions for Authors
This study investigates a novel DC-based cancer immunotherapy enhanced by chitosan nanoparticles. The findings demonstrate that DC-CH-NPs significantly enhance DC maturation and T-cell activation, highlighting their potential as an effective cancer vaccine. The study focuses on DC maturation and CD4+ T-cell activation, but does not investigate CD8+ T-cell responses, which are crucial in tumor immunity. A discussion on its potential role would be beneficial.
Specifc comments
1, Abstract (Lines 24-25): "Spleen samples also expressed DC markers, particularly CD83."
Suggested revision: "DC markers, particularly CD83, were highly expressed in spleen samples."
2, Methods (Lines 140-142): "The formed CH-NPs are not intended to be a gel since vaccination is intramuscular."
Suggested revision: "The formed CH-NPs were designed to remain in a nanoparticle suspension to ensure effective intramuscular vaccination."
3, Discussion (Lines 578-579): "CH-NPs are considered the most extensively studied nanosystem in drug research." This statement is too broad. Suggested revision: "CH-NPs are widely studied in drug delivery research due to their biocompatibility and mucoadhesive properties."
Comments on the Quality of English LanguageThe overall English proficiency needs improvement. I recommend a thorough language revision to enhance clarity and readability.
Author Response
We sincerely thank the reviewers for their thoughtful and constructive feedback on our manuscript "Impact of Chitosan Nanoparticles Coated Dendritic Cells-Based Vaccine as Cancer Immunotherapy." We greatly appreciate the time and effort invested in evaluating our work. The comments provided have been invaluable in improving our study's clarity, depth, and scientific rigor. We have carefully considered all suggestions and have revised the manuscript accordingly. Below, we provide a point-by-point response outlining the changes made and the sections where they have been incorporated.
Comments and Suggestions for Authors
This study investigates a novel DC-based cancer immunotherapy enhanced by chitosan nanoparticles. The findings demonstrate that DC-CH-NPs significantly enhance DC maturation and T-cell activation, highlighting their potential as an effective cancer vaccine. The study focuses on DC maturation and CD4+ T-cell activation, but does not investigate CD8+ T-cell responses, which are crucial in tumor immunity. A discussion on its potential role would be beneficial.
Response:
Thank you for this valuable suggestion. We agree that CD8⁺ T-cell responses are critical for antitumor immunity. We have now included a paragraph at the end of the Discussion Section (Lines 532- 546) discussing the importance of CD8⁺ T cells and explaining their omission in this preliminary study due to reagent and panel limitations, with a commitment to investigate their role in future studies.
Specific comments
1, Abstract (Lines 24-25): "Spleen samples also expressed DC markers, particularly CD83."
Suggested revision: "DC markers, particularly CD83, were highly expressed in spleen samples."
Response:
Thank you for this valuable suggestion. Modification is done (lines 27-28)
2, Methods (Lines 140-142): "The formed CH-NPs are not intended to be a gel since vaccination is intramuscular."
Suggested revision: "The formed CH-NPs were designed to remain in a nanoparticle suspension to ensure effective intramuscular vaccination."
Response:
Thank you for this valuable suggestion. Modification is done (lines 87-88).
3, Discussion (Lines 578-579): "CH-NPs are considered the most extensively studied nanosystem in drug research." This statement is too broad. Suggested revision: "CH-NPs are widely studied in drug delivery research due to their biocompatibility and mucoadhesive properties."
Response:
We agree that the original statement was too general. The sentence has been revised accordingly to highlight their specific advantages in drug delivery (lines 510-511).
Comments on the Quality of the English Language
The overall English proficiency needs improvement. I recommend a thorough language revision to enhance clarity and readability.
Response:
We acknowledge this, and the manuscript has been thoroughly revised for clarity, grammar, and scientific tone. Major improvements were made in the Abstract, Introduction, Results, and Discussion sections.
Reviewer 2 Report
Comments and Suggestions for Authors
The manuscript of J.S. Alrahimi entitled “Impact of Chitosan nanoparticles coated dendritic cells-based vaccine as cancer immunotherary” deals with a very important topic. The section on the functionality of immunotherapy is well described; however, the entire part concerning the development of nanoparticles and their combination with dendritic cells would need significant improvement. There is a lack of a detailed description of the methodology used for the preparation of the nanoparticles, as well as a need for thorough characterization. Information on the morphology of the nanoparticles, their surface charge, and the nature of the interaction between the nanoparticles and the dendritic cells is missing. Additionally, the term 'coating' is used incorrectly throughout the manuscript. If the nanoparticle is pre-formed before mixing with the dendritic cell, the cationic charge from the chitosan would interact with the cell surface, but it is not explained whether it fully covers the dendritic cell. There are also redundancies in the Materials and Methods section—specifically in sections 2.1.1, 2.1.2, 2.4, and 2.5.
Regarding Figure 4, the particle size distribution shows that the sample is highly heterogeneous, with one population of particles around 3 nm (which likely are not the chitosan nanoparticles) and another population around 60 nm. The average particle size should be reported in the main text, not just in Figure 4. It would be helpful for the authors to include the mean particle diameter provided by the equipment, as well as the polydispersity index of the measurement. Furthermore, the FT-IR spectra do not provide useful information. The amino and -OH groups of chitosan are visible, but these peaks are always present in chitosan alone and also when crosslinked with TPP. These measurements do not confirm the formation of nanoparticles as the authors suggest, and even less so the interaction with dendritic cells.
All of these aspects must be thoroughly described before this manuscript can be considered for publication.
Author Response
We sincerely thank the reviewers for their thoughtful and constructive feedback on our manuscript "Impact of Chitosan Nanoparticles Coated Dendritic Cells-Based Vaccine as Cancer Immunotherapy." We greatly appreciate the time and effort invested in evaluating our work. The comments provided have been invaluable in improving our study's clarity, depth, and scientific rigor. We have carefully considered all suggestions and have revised the manuscript accordingly. Below, we provide a point-by-point response outlining the changes made and the sections where they have been incorporated.
Comments and Suggestions for Authors
The manuscript of J.S. Alrahimi entitled “Impact of Chitosan nanoparticles coated dendritic cells-based vaccine as cancer immunotherary” deals with a very important topic. The section on the functionality of immunotherapy is well described; however, the entire part concerning the development of nanoparticles and their combination with dendritic cells would need significant improvement. There is a lack of a detailed description of the methodology used for the preparation of the nanoparticles, as well as a need for thorough characterization.
Response
We thank the reviewer for this observation. We have revised the manuscript to provide a more comprehensive and detailed description of the methodology used for the preparation of chitosan nanoparticles (CH-NPs). Specifically, we have clarified the concentrations, preparation steps, mixing conditions, and ionic gelation process, including the role of tripolyphosphate (TPP) as a crosslinking agent. These revisions have been made in Section 2.2.1: Preparation of Chitosan Nanoparticles to enhance reproducibility and transparency (lines 83-92).
Additionally, we have added a sentence in Section 2.1.1 explaining that CH-NPs interact with dendritic cell membranes via electrostatic attraction, due to their positive surface charge and the negative charge of cell membranes. This further supports the rationale behind the nanoparticle preparation and its intended immunological function (lines 89-92).
We hope that these improvements adequately address the reviewer’s concern and strengthen the methodological clarity of our study.
Information on the morphology of the nanoparticles, their surface charge, and the nature of the interaction between the nanoparticles and the dendritic cells is missing. Additionally, the term 'coating' is used incorrectly throughout the manuscript. If the nanoparticle is pre-formed before mixing with the dendritic cell, the cationic charge from the chitosan would interact with the cell surface, but it is not explained whether it fully covers the dendritic cell.
Response:
Thank you for highlighting this. We have revised the manuscript to more accurately describe the nanoparticle-cell interface as “surface adsorption of CH-NPs onto DC membranes” rather than "coating." Additionally, we have expanded the methodology to describe the nanoparticle preparation process and ionic interaction mechanism, and clarified the terminology (Section 2.1.1, lines 87-90). The title of section 2.4 ( line 151), section 2.4 (lines 152-154).
There are also redundancies in the Materials and Methods section—specifically in sections 2.1.1, 2.1.2, 2.4, and 2.5.
Response:
We appreciate the observation and have merged repetitive content between Sections 2.1.1 (Preparation) and 2.4 (Nanoparticle Preparation), and likewise between 2.1.2 (Characterization) and 2.5 (Characterization). These sections now flow more concisely without loss of information.
Regarding Figure 4, the particle size distribution shows that the sample is highly heterogeneous, with one population of particles around 3 nm (which likely are not the chitosan nanoparticles) and another population around 60 nm. The average particle size should be reported in the main text, not just in Figure 4. It would be helpful for the authors to include the mean particle diameter provided by the equipment, as well as the polydispersity index of the measurement. Furthermore, the FT-IR spectra do not provide useful information. The amino and -OH groups of chitosan are visible, but these peaks are always present in chitosan alone and also when crosslinked with TPP. These measurements do not confirm the formation of nanoparticles as the authors suggest, and even less so the interaction with dendritic cells.
Response:
We thank the reviewer for this detailed and helpful comment. We agree that FT-IR spectra alone do not confirm the formation of nanoparticles or their interaction with dendritic cells, as the observed –NH₂ and –OH peaks are indeed characteristic of chitosan, regardless of its state (polymer or nanoparticle). Accordingly, we have revised the interpretation in the main text (Section 3.3) to clarify that FT-IR was used to verify the presence of functional groups and assess structural consistency, but not to confirm nanoparticle synthesis (lines 276- 281).
In response to the reviewer’s suggestion, we have now included the mean particle diameter (65 nm) and polydispersity index (PDI = 0.224) as measured by DLS in the main text of Section 3.3, and we have revised the caption of Figure 4 to reflect this quantitative information (lines 295- 297). We have also clarified that the minor population around 3 nm is likely background noise or related to solvent components and does not represent CH-NPs (lines 284-288).
These modifications ensure that the evidence for nanoparticle formation is based on particle size distribution, surface charge (zeta potential), and biological functionality rather than FT-IR alone (the Discussion section, lines 519- 523).
We sincerely thank the reviewers for their valuable time, insightful feedback, and constructive suggestions, which have significantly improved our manuscript's clarity, rigor, and scientific quality. We have carefully addressed each comment and made the necessary revisions accordingly. We greatly appreciate the opportunity to revise our work and are confident that the updated manuscript is stronger and more comprehensive.
Round 2
Reviewer 2 Report
Comments and Suggestions for Authors
The article has been significantly improved in its description of the preparation and characterization of the nanoparticles; therefore, I consider it suitable for acceptance.